# Production of Plant Beneficial and Antioxidants Metabolites by *Klebsiella*
*variicola* under Salinity Stress

**DOI:** 10.3390/molecules26071894

**Published:** 2021-03-26

**Authors:** Supriya P. Kusale, Yasmin C. Attar, R. Z. Sayyed, Roslinda A. Malek, Noshin Ilyas, Ni Luh Suriani, Naeem Khan, Hesham A. El Enshasy

**Affiliations:** 1Department of Microbiology, Rajaram College, Kolhapur 416004, India; supriya.kusale@gmail.com; 2Department of Microbiology, P.S.G.V.P. Mandal’s, Arts, Science, and Commerce College, Shahada 425409, India; sayyedrz@gmail.com; 3Institute of Bioproduct Development (IBD), Universiti Teknologi Malaysia (UTM), Skudai 81310, Malaysia; roslinda@ibd.utm.my; 4Department of Botany, PMAS Arid Agriculture University, Rawalpindi 46300, Pakistan; noshinilyas@yahoo.com; 5Biology Department, Faculty of Mathematics and Natural Science, Udayana University, Bali 80361, Indonesia; niluhsuriani@unud.ac.id; 6Department of Agronomy, Institute of Food and Agricultural Sciences, University of Florida, Gainesville, FL 32611, USA; naeemkhan@ufl.edu; 7City of Scientific Research and Technology Applications (SRTA), New Burg Al Arab, Alexandria 21934, Egypt

**Keywords:** ACC deaminase, antioxidant enzymes, PGP traits, phytase, salinity stress

## Abstract

Bacteria that surround plant roots and exert beneficial effects on plant growth are known as plant growth-promoting rhizobacteria (PGPR). In addition to the plant growth-promotion, PGPR also imparts resistance against salinity and oxidative stress and needs to be studied. Such PGPR can function as dynamic bioinoculants under salinity conditions. The present study reports the isolation of phytase positive multifarious *Klebsiella variicola* SURYA6 isolated from wheat rhizosphere in Kolhapur, India. The isolate produced various plant growth-promoting (PGP), salinity ameliorating, and antioxidant traits. It produced organic acid, yielded a higher phosphorous solubilization index (9.3), maximum phytase activity (376.67 ± 2.77 U/mL), and copious amounts of siderophore (79.0%). The isolate also produced salt ameliorating traits such as indole acetic acid (78.45 ± 1.9 µg/mL), 1 aminocyclopropane-1-carboxylate deaminase (0.991 M/mg/h), and exopolysaccharides (32.2 ± 1.2 g/L). In addition to these, the isolate also produced higher activities of antioxidant enzymes like superoxide dismutase (13.86 IU/mg protein), catalase (0.053 IU/mg protein), and glutathione oxidase (22.12 µg/mg protein) at various salt levels. The isolate exhibited optimum growth and maximum secretion of these metabolites during the log-phase growth. It exhibited sensitivity to a wide range of antibiotics and did not produce hemolysis on blood agar, indicative of its non-pathogenic nature. The potential of *K. variicola* to produce copious amounts of various PGP, salt ameliorating, and antioxidant metabolites make it a potential bioinoculant for salinity stress management.

## 1. Introduction

The excess use of agrochemicals has resulted in the depletion of nutrients and useful rhizobia from soil [1]. The continued use of these chemicals has also severely affected soil fertility, soil health, agro-ecosystem, and human health [2]. These negative impacts of agrochemicals highlight the need for sustainable substitutes [3]. The salinity of agricultural soil, i.e., the presence of excess amounts of salts, is one of the major abiotic stresses [4,5]. Soil salinity is an increasing problem worldwide for about 27% of the world’s arable land [6]. Plant growth in these soils is adversely affected by both excess salts and excess sodium levels [7]. Saline soil prevents the normal growth of crops and results in poor crop yield [7]. The effects of soil salinization are more detrimental, particularly in arid and semi-arid regions.

The use of organic manure and farmyard manure has been successful in restoring soil health and soil fertility, however, these are less effective and not sustainable [4]. The various conventional methods of ameliorating soil salinity have provided some success, but they show negative impacts on soil health [6]. A sustainable agriculture practice demands effective and eco-friendly measures to combat these issues without compromising soil health [7]. In this context, the application of plant growth-promoting rhizobacteria (PGPR) has proved the best alternatives to agrochemicals and effective bioinoculants to promote plant growth in saline soils [1]. PGPR exerts a spectrum of beneficial effects on crop plants, including plant growth promotion and the amelioration of soil salinity and antioxidant activities [8]. PGPR helps plant growth through phosphate, iron, and other mineral nutrition, provision of nitrogen, ammonia, and phytohormones [1]. Aminocyclopropane-1-carboxylate deaminase (ACCD) produced by PGPR promotes root growth [6]. PGPR also excrete exopolysaccharides (EPS) which help in salinity amelioration [8]. Applications of plant growth-promoting rhizobacteria (PGPR) are known as effective, and sustainable inputs for the replenishment of soil nutrients, plant nutrients, and plant health [9]. Various reports claim that the application of PGPR for sustainable improvements in nutrients and the health of soil and plants [4,9,10,11]. PGPR are known to produce a wide variety of plant growth-promoting traits that include siderophores [10,11], phytohormones [12,13], hydrolytic enzymes [14], exopolysaccharide [15,16,17], stress-tolerant metabolites [18,19,20,21], and phosphate solubilization [22,23], etc. PGPR through these metabolites promotes seed germination, root and shoot growth, the level of soil and plant nutrients [13,21,24]. The beneficial effects of PGPR are sometimes not obtained due to various reasons such as the production of a limited number of plant growth-promoting (PGP) traits, competition with the native soil microflora, less root colonization, and sensitivity to biotic and abiotic stresses [2,3,20]. A multifarious and stress-tolerant PGPR can serve as an effective bioinoculant and stress elevator [25] that also helps in restoring soil health [26,27]. To be used as an effective bioinoculant, it is desired to confirm the multifarious plant growth-promoting, salinity ameliorating, and antioxidant ability of such PGPR. Therefore, the present study was aimed to screen multifarious PGPR with the potential of producing, PGP, salinity ameliorating, and antioxidant metabolites. The present study reports the isolation of *Klebsiella variicola* SURYA6 isolated from wheat rhizosphere and the production of multiple PGP, salinity ameliorating, and antioxidant traits.

## 2. Results

### 2.1. Production of Plant Growth-Promoting Traits

#### 2.1.1. Screening for Phytase Production

A total of 34 bacterial isolates were obtained from the collected soil samples. Four isolates, namely N6, H7, B1, and V8 produced larger zones of P solubilization among the other isolates, hence they were selected for further studies. The isolate N6 yielded maximum P solubilization index (9.3 mm) and maximum phytase activity (376.67 ± 2.77 U/mL) vis-à-vis other isolates (Table 1). Among the 34 isolates, four isolates showed a clear zone on PVK and KB agar plates. The isolate N6 showed maximum P solubilization on PVK agar and in PVK broth (Table 1).

NA plates supplemented with the pH indicator dye and separately inoculated with four isolates, namely N6, H7, B1, and V8, showed a change in the color of the medium from yellow to pink, surrounding the colonies of these isolates. This color change in the color of colonies indicated the production of organic acids. Isolate N6 showed dark, intense pink color as compared to the other isolates.

#### 2.1.2. Screening for Nitrogen Fixation, Production of Ammonia and Siderophore

All the isolates were found to grow luxuriously on nitrogen-deficient MM (Table 1) indicating their nitrogen fixation ability. All the isolates produced ammonia and siderophore in varying amounts, however, the isolate N6 produced maximum amounts of ammonia and siderophore units (79.0%) as compared to the other isolates (Table 1), and hence it was considered as multifarious PGPR.

### 2.2. Production of Salinity Ameliorating Traits

The isolate N6 produced the maximum amount of IAA, ACCD, and EPS compared to the H7, B1, and V8 isolates (Table 1).

### 2.3. Screening for Antioxidant Enzymes

Among the various isolates, isolate N6 produced higher activities of SOD (13.86 IU/mg protein), CAT (0.053 IU/mg protein), and GSH (22.12 µg/mg protein) (Table 1).

### 2.4. Effect of Incubation Period on Growth, and Production of PGP and Antioxidant Traits

Growth phases of isolate N6 showed a lag phase of 4 h, an active exponential phase from 6 to 30 h, a stationary phase after 30 h, and a decline phase from 36 h onwards. Production of PGP and antioxidant traits began from 12 h and continued up to 48. The maximum amounts of PGP traits and antioxidant enzymes were produced during the end of the exponential phase (30 h). A significant decline in the amounts of these traits occurred after 30 h growth. The isolate N6 produced a maximum phytase activity of 8.46 ± 2.1 (IU/mL) (Figure 1A); optimum siderophore yield (77.42 ± 1.9 SU); (Figure 1B), IAA (78.45 ± 1.9 µg/mL) (Figure 1C), maximum EPS yield of 32.2 ± 1.2 g/L (Figure 1D) and highest ACCD activity (0.891 M/mg^/^h) (Figure 1E) during 30 h of incubation.

The highest activities of antioxidant enzymes were also recorded during the log phase (30 h) of growth. The isolate exhibited optimum CAT activity, i.e., 0.81 IU/mg protein during 24 h of growth; (Figure 2A); while it produced the highest SOD activity of 14.96 IU/mg protein (Figure 2B); and the maximum GSH oxidase activity of 34.09 IU/mg protein (Figure 2C); during 36 h of incubation (Figure 2C).

### 2.5. Effect of Salt Concentration on Growth and Production of PGP Taits and Antioxidant Enzymes in K. variicola SURYA6

*K. variicola* produced a good amount of PG traits at a varying range of salt concentrations. An increase in the amounts of PGP traits was observed with increasing concentrations of salt. The threshold level of salt (NaCl) that inhibited phytase activity was 100 mM (Figure 3). Siderophore production (Figure 3B) ceased and was negatively impacted above 100 mM of NaCl concentration, while salt-ameliorating traits such IAA (Figure 3C), EPS (Figure 3D), and ACCD (Figure 3D) reflected a higher threshold level (120 mM) of salt. The maximum phytase activity of 8.46 ± 2.1 (IU/mL) was observed at 120 mM of NaCl (Figure 3A); the optimum siderophore units (77.42 + 0.18) were produced at 100 mM of NaCl (Figure 3B) while the highest amount of IAA (99.32 ± 1.8 µg/mL) (Figure 3C), maximum EPS (72.2 ± 1.1 g/L) (Figure 3D) and optimum ACCD activity (0.981 M/mg^/^h) (Figure 3E) were recorded at 120 mM of salt.

The isolate N6 produced the antioxidant enzymes at all the concentrations of salt, however, NaCl concentration above 100 mM significantly affected the production of these enzymes. The optimum CAT activity, i.e., 0.55 ± 1.7 IU/mg protein (Figure 4A); SOD activity of 82.21 ± 1.9 IU/mg protein (Figure 4B); and maximum GSH oxidase activity of 35.09 ± 1.8 IU/mg protein (Figure 4C); were obtained at 100 mM salt (NaCl) concentration.

### 2.6. Identification of the Potent Isolate-Ribotyping

Comparison of 16S rRNA gene sequence analysis of isolate N6 exhibited 99.82% similarity with *Klebsiella variicola* (Figure 5), thus the isolate was identified as *Klebsiella variicola*, and 16S rRNA gene sequences of the isolate were deposited in the National Center for Biotechnology Information (NCBI) gene bank under the name *Klebsiella variicola* SURYA6 with the accession number MF187615.1.

## 3. Discussion

Agricultural soil is a rich source of a diverse population of PGPR [6,9]. These PGPR provide several benefits to the plants such as plant growth promotion, the enrichment of plant nutrients, as well the enrichment of the nutrients in soil [1]. They exert a wide variety of plant beneficial activities such as N_2_ fixation, the production of IAA, ammonia, HCN, siderophore, EPS, and P solubilization [5,12]. In addition to plant growth-promoting effects, PGPR produces a wide range of metabolites that protect the plant from oxidative damages caused due to the excess of salts [16,17,18].

The phytase-producing soil bacteria widely occur in the rhizosphere of different plant species [26,27,28,29,30]. Phytase-producing *K. variicola* sp. has been isolated as endophytes from banana and maize leaves [31,32]. The isolate produced phytase and yielded a 39 mm clear zone of phytate solubilization. Singh et al. and Liu et al. reported the isolation of phytase producing *Bacillus subtilis* strain DR6 and *Bacillus* sp., respectively, from maize rhizosphere, they reported the 378 U/mL of phytase [29,30]. Reports of the present study on the phytase activity of *K. variicola* are in line with the reports of Singh et al. [28]. This study reports the highest phytate solubilization index than *K. pneumonia* isolated from the rhizosphere of sugarcane, and rice [31,32,33].

P solubilization is attributed to the secretion of phytase and the production of organic acids that decrease the pH of the medium. The secretion of phytase has been reported as the key factor in P solubilization [34,35]. The production of IAA is regarded as the principal criterion for screening PGPR because IAA-producing bacteria are known to exert profound beneficial effects on plant growth [6]. The production of ammonia has been reported in many P solubilizing *Klebsiella* sp. isolated from the various plant rhizospheres [33,34].

The production of ammonia is a well-known mechanism of PGPR adopted for plant growth promotion [35,36]. Bhardwaj et al. [30] isolated phytase positive, P solubilizing, and IAA producing multifarious *K. pneumonia* from the rhizosphere of sugar cane. Kuan et al. reported two sp. of *Klebsiella* namely *Klebsiella* sp. Br1 and *K. pneumonae* Fr1 to fix N_2_, solubilize P, and produce IAA [37]. Sachdev et al. reported the production of IAA in six strains of *K. pneumoniae* isolated from the wheat rhizosphere [38]. Mazumdar et al. isolated P solubilizing *K. pneumoniae* rs26 from the chickpea rhizosphere. They also reported N_2_ fixation and IAA production in this isolate [33]. Rosenblueth et al., reported N_2_ fixation activity in *K. variicola* [30]. Govindarajan et al. isolated *Klebsiella* sp. GR9 from sugarcane rhizosphere and found the bacterium capable of producing multiple plant growth-promoting traits [39].

Sayyed et al., reported siderophore production in *Achromobacter* sp. isolated from d rhizosphere [9]. Shaikh et al. observed siderophore production in *Pseudomonas* sp. isolated from the banana rhizosphere [8]. Haahtela et al. reported enterochellin siderophore production in *K. pneumoniae* and *K. terrigena* isolated from plants [40]. These isolates also produced auxins and IAA. Marasco et al. reported multifarious *Klebsiella* sp. from pepper rhizosphere [41].

EPS production has been reported in various members of *Klebsiella* sp. such as *K. oxytoca* [42], *K. pnuemoniae* [42,43], *K. pneumoniae* sp. *pneumoniae* serotype K63 [44], and *Klebsiella* sp. PHRC1.001 [45]. The production of EPS by rhizobacteria is one of the useful traits, as it chelates excess salt ions, thus protecting the plant from the osmotic effects of salts [45,46]. *K. variicola* SURYA6 exhibited higher yields (32.2 ± 1.2 g/L) of vis-à-vis *K. oxytoca* (6–15 g/L) reported by Hariyono [47], and *K. pneumoniae* strain DSM 30,104 (1.75 mg/mL) reported by Dlamini et al. [48].

The ability of the isolate to grow in high concentrations of various salts can be attributed to the excretion of EPS that maintains the viability of bacterial cells under salt stress and protects them in the rhizosphere [6]. Good growth of the isolate at higher concentrations of a wide variety of salts indicates its halophilic physiology and its ability to survive under salinity stress. Halophilic organisms grow at higher salt levels (100–300 mM) as they can maintain the balance of the osmotic pressure of the environment and thus protect their enzymes and proteins [49]. Halophilic *Klebsiella* sp. isolated from the plant roots showed tolerance to higher levels (4 to 8% NaCl) of salts [22]. Singh et al., reported the growth of *Klebsiella* sp. at a higher level (6% of NaCl (Salt) [50]. The results of the present studies are in line with Arora et al. [51].

The majority of the PGP and salt-ameliorating traits are produced during the active/log-phase growth of PGPR [52]. Sayyed et al. and Reshma et al. reported the production of EPS, siderophore, and antioxidant enzymes during the exponential growth of *Enterobacter* sp. RZS5, *Achromobacter* sp., and *Pseudomonas* sp., respectively [9,15,53]. The reduction in growth and decreased synthesis of PGP metabolites and antioxidant enzymes is because of the reduced availability of nutrients, accumulation of toxic wastes, and unfavorable changes in the pH of the medium [18]. Rhizobacteria are known to ameliorate salinity stress through the production of ACCD [17,18]. Halophiles of Enterobacteriaceae sp. that are adapted to salinity stress produce an array of PGP metabolites [54] like siderophores [9,15], exopolysaccharides (EPS), phytohormones, and various stress-tolerant enzymes [53]. Sagar et al., reported the production of various PGP traits and ACCD in *E.cloacae* PR4 [16]. Jabborova et al., reported the production of various PGPR traits in halophilic endophytes [11].

Production of ACCD is attributed to the presence of ACC in MM [18] and the production of ACCD in PGPR is one of the best mechanisms of salinity tolerance [9]. The enzyme ACCD lowers the level of ACC in root exudates, the suboptimal level of ACC reduces the concentration of ethylene in the plant roots and thus helps in root length which improves the absorption of nutrients [9]. Srivastav and Kumar reported the 0.539 µM/mg/h activity of ACCD in *Klebsiella* sp. strain ECI-10A [55]. Singh et al. reported ACCD and other plant-growth-promoting activities in *Klebsiella* sp. SBP-8 isolated from *Sorghum bicolor* from the desert region of Rajasthan, India [50]. The isolate exhibited optimum growth and ACCD activity at high salt (NaCl) concentrations of up to 6%. The isolate ameliorated 150–200 mM of salt in wheat and promoted plant growth under salinity conditions. ACCD-producing rhizobacteria are widely studied as PGPR to elevate a variety of stresses in plants [56]. Acuña et al., isolated *Klebsiella* isolate sp. 8LJA and 27IJA from *Parastrephia quadrangularis rhizosphere* [56]. These isolates produced ACCD and SOD activities under salt stress (250 mM and 450 mM NaCl). The presence of ACCD and antioxidant enzyme-producing PGPR activates a defensive system in the plants which removes the free radicals produced due to salt or other stresses [57].

Antioxidant enzymes like SOD, CAT, and GSH produced by PGPR protect plants from oxidation due to osmotic shocks [3]. Sapre et al. isolated halophilic *Klebsiella* sp. IG 3 from *Avena sativa* rhizosphere [58]. Their isolate tolerated salinity stress up to 100 mM, produced IAA, proline, and antioxidant enzymes under normal and salinity stress conditions, and helped in salinity stress amelioration and growth promotion in *A. sativa*. The production of antioxidant enzymes in *K. variicola* is due to its halophilic nature. To prevent the oxidation of their important biomolecules to protect the cell from the osmotic damage of salts, halophilic microorganisms produce antioxidant enzymes [3]. Salinity conditions create oxidative stress that damages the cell membranes and cell structures in microbes as well as plant cells. Under salinity stress conditions, the presence of antioxidant enzyme-producing PGPR activates an antioxidative defensive mechanism in the plants and removes the free radicals produced due to salinity stress [57].

Multiple resistances to a wide variety of antibiotics are commonly exhibited by *Klebsiella* sp. associated with clinical infections [31]. However, the isolate, *K. variicola* SURYA6, exhibited sensitivity towards 26 different types of antibiotics; moreover, the inability of the isolate to grow on BA confirmed its non-clinical and non-pathogenic origin. Some reports showed that the *K. variicola* is associated with extended-spectrum beta-lactamase (ESBL)-producing ability, however, *K. variicola* SURYA6 did not show ESBL activity. Barrios et al. [25] claimed *K. variicola* was a versatile bacterium capable of colonizing different hosts such as plants, humans, insects, and animals. *K. variicola* of plant origin is non-pathogenic to humans or animals [59].

The ability of the isolate to grow under salinity stress through the secretion of salinity ameliorating and antioxidant metabolites and to secrete multiple plant growth-promoting characters makes it a good bioinoculant for plant growth promotion while removing the excess salts. However, multiple field studies over the period and under different agroclimatic conditions need to be explored to use this isolate as a bioinoculant at the commercial level.

## 4. Material and Methods

### 4.1. Soil Samples

Ten grams of rhizosphere soils of wheat (*Triticum aestivum* L.) were collected each from four agroecological zones, namely the Valivade (16.7185° N, 74.3133° E), Hatakanagale (16.7446° N, 74.4467° E), Nandani (16.7262° N, 74.5434° E), and Balinga (16.6878° N, 74.1703° E) villages of Kolhapur District of Maharashtra, India.

### 4.2. Screening for Multifarious PGPR Isolate

#### 4.2.1. Screening for Phytase Production

All the soil samples were individually mixed @ 1.0 g in separate sterile saline, serially diluted and 0.1 mL of 10^−6^ aliquots of each sample were individually incubated in sterile nutrient broth (NB) at 28 ± 2 °C for 48 h. A 0.1 mL of culture broth from each NB was separately spread on phytase screening medium [60] and incubated at 28 ± 2 °C for 48 h and then observed for the zone of hydrolysis of phytate around the colonies as an indication of extracellular phytase production. An un-inoculated phytase screening medium served as a control. The efficient phytase producers were selected based on the phytate solubilizing index (PSI). PSI is calculated as follows:(1)$$P.S.I\;=\frac{\mathrm{Colony}\;\mathrm{diameter}+\mathrm{Zone}\;\mathrm{diameter}}{\mathrm{Colony}\;\mathrm{diameter}}$$

For phytase assay, each isolate was individually grown in Pikovskaya’s (PKV) broth [61] at 28 ± 2 °C for 48–72 h, and then the broth was centrifuged at 10,000 rpm for 10 min. After that, the phytase activity of the supernatant was measured according to the method of Fiske and Subbarao [62]. One unit of phytase was defined as the amount of phytase that liberates 1 µM of inorganic phytate per min at 28 °C at pH 6.5.

#### 4.2.2. Phosphate Solubilization and Organic Acid Production

P solubilization ability of isolates was checked on Pikovskaya’s (PKV) agar [61], and Katznalson’s and Bose (KB) agar [63]. For this purpose, the log phase cultures (5 × 10^−5^ cell/mL) of each isolate from phytase screening medium were separately grown on PKV and KB agar at 28 °C for 24–48 h, and then observed for the formation of P solubilization around the colonies. Uninoculated PKV and KB agar served as a control. The isolates that formed bigger zones of P solubilization were selected as efficient P solubilizing bacteria (PSB) [61].

For the quantitative estimation of P solubilization, log-phase cultures (5 × 10^−5^ cells/mL) of each potent PSB were individually grown in each PKV broth at 28 ± 2 °C and 120 rpm for 72 h, and then the broth was centrifuged at 10,000 rpm for 10 min. The inorganic P in the cell-free supernatant was determined according to the method of Fiske and Subbarao [62]. An un-inoculated PKV broth served as a control.

Screening for organic acid production was performed on nutrient agar nutrient agar (NA) supplemented with 1% sugar and 0.05% *(w/v)* of acid fuchsine (pH indicator dye). Each culture was separately grown on NA at 28 ± 2 °C for 24 h and observed for change in the color of the medium from pale yellow to dark pink [64].

#### 4.2.3. Screening for Nitrogen Fixation and Production of Ammonia, Siderophore and Indole Acetic Acid (IAA)

For screening N_2_ fixation ability, log-phase cultures (5 × 10^−5^ cells/mL) of each isolate were grown in the nitrogen-deficient mineral media (NDM) at 28 ± 2 °C for 24 h and observed for the occurrence of bacterial growth. The occurrence of the growth of isolate on NDM was taken as an indication of the N_2_ fixing ability of the isolate [65]. Uninoculated NDM served as a control.

For the screening of ammonia production, log phase culture (5 × 10^−5^ cells/mL) of each isolate was grown in the sterile peptone water (PW) medium at 28 ± 2 °C for 24 h, and plates were observed for the occurrence of yellow color as a sign of ammonia production [66]. Uninoculated PW medium served as a control.

For the evaluation of siderophore-producing ability, log phase culture (5 × 10^−5^ cells/mL) of each isolate was individually grown on Chrome Azurol S (CAS) agar plates at 28 ± 2 °C for 24–48 h. Following the incubation, plates were observed for the formation of yellow-orange halos surrounding the colonies [67]. Uninoculated CAS agar served as a control.

For siderophore production, log phase cultures (5 × 10^5^ cells/mL) of each isolate was separately grown in succinate medium [68] at 28 ± 2 °C for 24–48 h and the cell broth was centrifuged at 10,000 rpm for 10 min. Siderophore content (% siderophore units) from cell supernatant was estimated by CAS shuttle assay [69]. An uninoculated succinate medium was used as a reference.

For the screening of indole acetic acid (IAA) production, each isolate (5 × 10^5^ cells/mL) was grown in NB; supplemented with 0.2% of tryptophan at 28 ± 2 °C for 48 h at 120 rpm and the amount of IAA was measured according to the method of Gordon and Weber [70]. Uninoculated NB served as a reference.

### 4.3. Screening for Salinity Ameliorating Traits

#### 4.3.1. Production of ACCD and EPS

For the screening and estimation of ACCD activity, log phase cultures (5 × 10^−5^ cells/mL) of each isolate were grown in minimal medium (MM) containing (g/L), KH_2_PO_4_, 2; K_2_HPO_4_, 0.5; MgSO_4_, 0.2; glucose, 0.2, and (NH_4_)_2_SO_4_, 0.19, at 30 °C for 48 h and were then observed for the appearance of growth of the isolate [71]. Uninoculated MM was used as a reference. ACCD activity was estimated as per the Penrose and Glick method [72]. The ACCD activity was defined as the amount of α-keto-butyrate produced per mg of protein per h.

For EPS production, log phase cultures (5 × 10^5^ cells/mL) of each isolate were grown in 5% glucose basal medium at 28 ± 2 °C for 72 h at 120 rpm and the medium was centrifuged at 10,000 rpm for 20 min. The EPS was precipitated and extracted with cold ethanol, dried at 50 ± 2 °C until a constant weight was obtained for the estimation of EPS [14,73].

#### 4.3.2. Screening for Production of Antioxidant Enzymes

For the screening for superoxide dismutase (SOD), catalase (CAT), and reduced glutathione oxidase (GSH) production, each isolate was separately grown in MM at 28 ± 2 °C for 30 h at 120 rpm. After incubation, the broth was centrifuged at 1000 rpm for 10 min. The uniform cell homogenate thus obtained was used for enzyme assay.

For the estimation of SOD activity, 100 µL of cell homogenate was mixed with 100 µL of pyrogallol solution in EDTA buffer (pH 7.0) and the absorbance was measured at 420 nm [74]. One unit of SOD was defined as the amount (IU/mg) of SOD required to inhibit 50% of the autoxidation of pyrogallol.

For measuring the catalase (CAT) activity, 100 µL of cell homogenate was mixed with 100 µL of hydrogen peroxide in phosphate buffer (pH 7.0), and the absorbance of the mixture was measured at 240 nm [75] by using a molar extinction coefficient of hydrogen peroxide (43.6 per M/cm). One unit of catalase was defined as an mM of hydrogen peroxide decomposed/min.

Reduced glutathione (GSH) activity was measured by mixing 100 µL cell homogenate in 100 µL of GSH and the absorbance of the mixture was measured at 240 nm [76]. GSH activity was measured as the reduction in µM of GSH per min.

### 4.4. Selection of Potent PGPR Isolate

The isolate that produced copious amounts of plant beneficial metabolites and salinity ameliorating traits was selected as a potent PGPR. The potent isolate was identified based on molecular characterization.

### 4.5. Growth Kinetics and Production of PGP and Salinity Ameliorating Traits in Potent PGPR

To decide the right time required for the optimum growth and production of PGP traits such as phytase, siderophore, IAA, ACCD, and EPS, the log phase culture (5 × 10^5^ cells/mL) of isolate N6 was separately grown in PKV broth, succinate medium Sabouraud’s broth, and MM, respectively, at 28 °C for 48 h. The samples were withdrawn after the 6 h interval and were assayed for the quantification of growth, phytase activity, the amount of siderophore, IAA, ACCD, and EPS [72,77,78,79].

To decide the exact time required for optimum cell growth and the optimum production of antioxidant enzymes, each isolate was separately grown in a basal medium at 28 °C for 48 h. Cell mass was harvested at 6 h intervals and assayed for the estimation of cell growth and SOD, CAT, and GSH activities [74,75,76].

### 4.6. The Effect of Salt Stress on the Production of PGP and Salinity Ameliorating Traits

The effect of various concentrations (0–200 mM) of salt (NaCl) on the growth and production of PGP and salt-ameliorating traits in the isolate N6 was evaluated by separately growing the log phase culture (5 × 10^5^ cells/mL) of isolate N6 in each nutrient broth (NB) containing varying amounts of NaCl salt (0–200 mM) at 28 °C for 48 h. Following incubation, cell growth was measured in terms of absorbance (optical density (OD) at 620 nm. For the estimation of PGP and salt-ameliorating traits, cell broth was centrifuged at 10,000 rpm for 20 min, then cell-free supernatant was assayed for the estimation of IAA, siderophore, EPS, activities of phytase, ACCD, SOD, CAT, and GSH as described above.

### 4.7. Identification of the Potent Isolate-Ribotyping

The phylogenetic identification of isolate N6 was performed by sequencing 16S rRNA genes of the isolate. The genomic DNA of N6 was isolated as per Sambrook and Russell [80]. The 16S rRNA genes were amplified by polymerase chain reaction (Elmer System 9700, Perkin, USA) by using the primers: 27f (5′-AGA GTT TGA TCC TGG CTC AG-3′) and 1492r (3′-ACG GCT ACC TTG TTA CGA CTT-5′) [81]. The 16S rRNA genes were amplified at denaturation at 94 °C for 5 min, 35 cycles of denaturation at 94 °C for 1 min, annealing at 55 °C for 1 min and extension at 72 °C for 1 min, final extension at 72 °C for 7 min with a final hold at 20 °C for infinity. The resulting PCR products were purified on agarose gel (1.0%) and sequenced on ABI 3730Xl automated sequencer using a ready reaction kit. The amplified sequences were analyzed by using gapped BLASTn. Phylogenetic analysis was performed using Molecular Evolutionary Genetics Analysis (MEGA) software version 8 [82] and phylogenetic trees were constructed [83]. The 16S rRNA gene sequences of the isolate were submitted to GenBank.

### 4.8. Statistical Analyses

All the experiments were performed in five replicates and the average values of five replicates. The effect of incubation and salt concentrations of the growth and production of metabolites was analyzed by one-way analysis of variance (ANOVA) followed by Turkey Honest Significant Difference (HSD) [84]. All statistical computations were performed on Statistica version 12.0 (Tibco Software, Palo Alto, CA, USA) and graphs were prepared by using origin (Origin Lab Corporation, Northampton, MA, USA).

## 5. Conclusions

The rhizosphere harbors a great and diverse abundance of PGPR. These PGPR produce various metabolites that are involved in plant growth promotion and tolerance to a wide range of biotic and abiotic stresses. Therefore, multifarious PGPR having the potential of growing under salinity conditions, the ability to secrete multiple PGP metabolites, salinity ameliorating traits, and an array of antioxidant enzymes can serve as the best PGPR as well as salinity elevator under field conditions. The potential of wheat rhizosphere *K. variicola* to grow over the range of high salt concentrations and produce copious amounts of multiple PGP metabolites, salt-ameliorating traits and antioxidant enzymes, make it a potential bioinoculant for plant growth promotion under saline soil. However, field trials under different seasons and different agroclimatic conditions can confirm the multiple potentials of *K. variicola*.

## Figures and Tables

**Figure 1 molecules-26-01894-f001:**
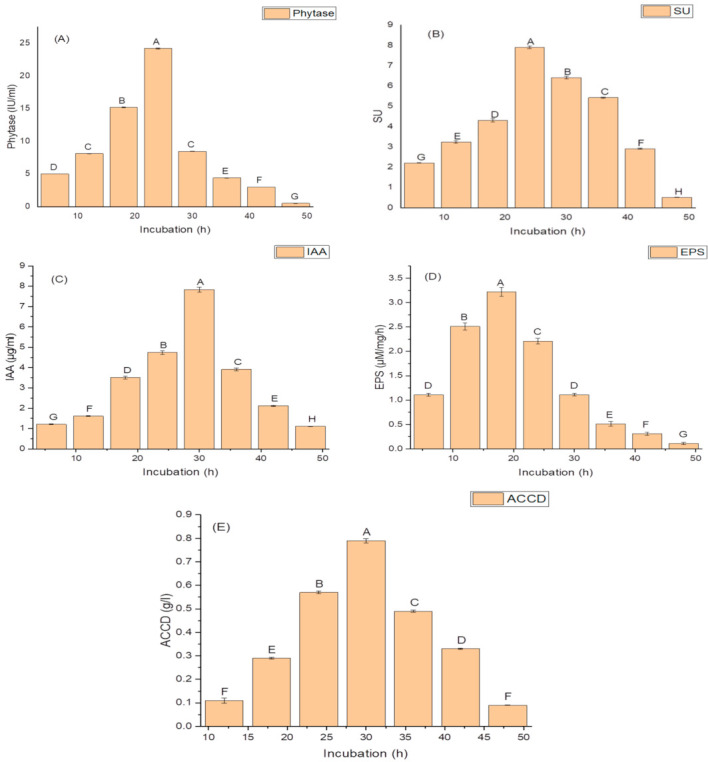
Effect of the incubation period on (**A**) phytase activity; (**B**) the production of siderophore units (SU); (**C**) Indole acetic acid (IAA) production; (**D**) Exopolysachharide (EPS) production; and (**E**) 1-aminocyclopropane-1-carboxylate deaminase (ACCD) activity of *K. variicola* SURYA6. Values are the average of five replicates and were analyzed by one-way ANOVA followed by Turkey’s test. Different letters on mean value of each parameter indicate significant differences at *p <* 0.05.

**Figure 2 molecules-26-01894-f002:**
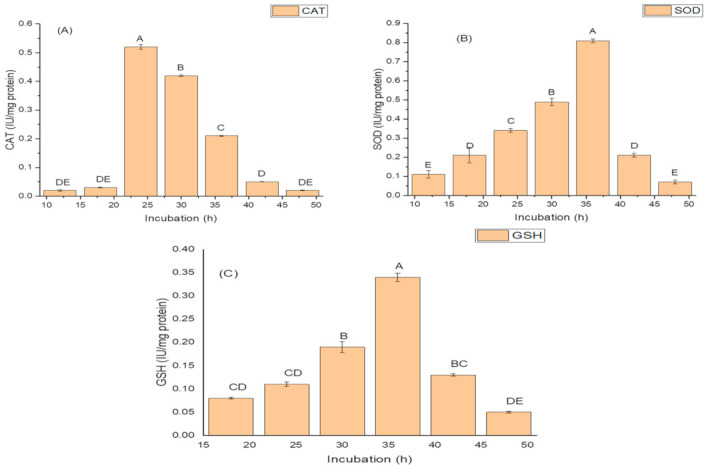
Effect of the incubation period on (**A**) catalase (CAT) activity; (**B**) superoxide dismutase (SOD) activity; and (**C**) glutathione reductase (GSH) activity of *K. variicola* SURYA6. Values are the average of five replicates and were analyzed by one-way ANOVA followed by Turkey’s test. Different letters on mean value of each parameter indicate significant differences at *p <* 0.05.

**Figure 3 molecules-26-01894-f003:**
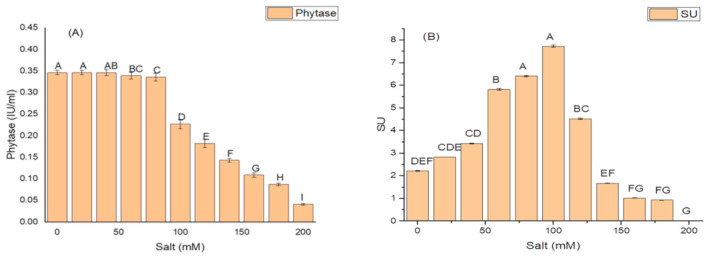
Effect of salt (NaCl) concentrations on (**A**) phytase activity; (**B**) production of siderophore units (Sus); (**C**) IAA production; (**D**) EPS production; and (**E**) ACCD activity of *K. variicola* SURYA6. Values are the average of five replicates and were analyzed by one-way ANOVA followed by Turkey’s test. Different letters on mean value of each parameter indicate significant differences at *p <* 0.05.

**Figure 4 molecules-26-01894-f004:**
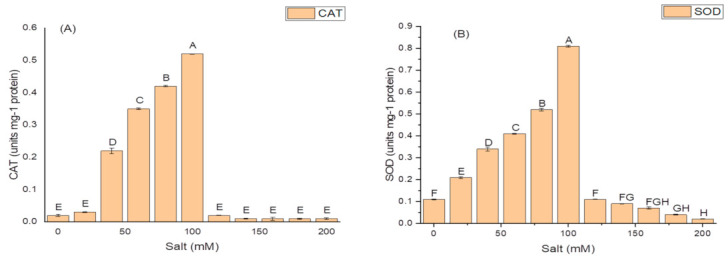
Effect of salt (NaCl) concentrations on (**A**) catalase (CAT) activity; (**B**) superoxide dismutase (SOD) activity; and (**C**) glutathione reductase (GSH) activity of *K. variicola* SURYA6. Values are the average of five replicates and were analyzed by one-way ANOVA followed by Turkey’s test. Different letters on mean value of each parameter indicate significant differences at *p <* 0.05.

**Figure 5 molecules-26-01894-f005:**
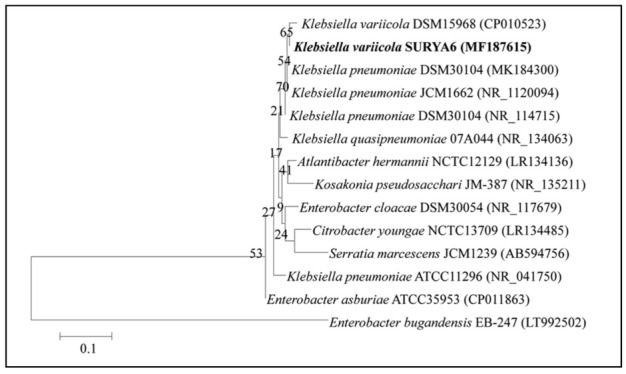
A phylogenetic tree showing the relatedness of the isolate N6 to other members of Genus *Klebsiella.* The 16S rRNA gene was amplified on PCR followed by electrophoresis. Amplified sequences were identified using the National Center for Biotechnology Information (NCBI) database and a phylogenetic tree was constructed.

**Table 1 molecules-26-01894-t001:** Screening for the production of PGP traits of various isolates.

Traits	Isolates			
	N6	H7	B1	V8
**Plant Growth-Promoting (PGP) Traits**				
Phytase activity (IU/mL)	346.67 ± 2.77	329.73 ± 1.33	261.67 ± 3.41	312.71 ± 2.11
P solubilization (µg/mL)	3229.02 ± 8.52	876.04 ± 3.21	507.01 ± 4.56	513.05 ± 6.78
Organic acid production	+++	+	+	+
Nitrogen fixation	+++	+	+	+
Ammonia production	+++	+	+	+
Siderophore production	79.0 ± 0.01	43.10 ± 0.01	52.11 ± 0.01	63.12 ± 0.01
IAA production (µg/mL)	78.45 ± 1.92	43.51 ± 3.91	5.22 ± 1.40	34.41 ± 1.7
**Salinity Ameliorating Traits**				
ACCD (µM/mg/h)	0.910 ± 1.21	0.782 ± 1.02	0.563 ± 1.01	0.312 ± 1.03
EPS production (g/L)	32.211 ± 1.21	8.12 ± 1.71	15.22 ± 1.91	23.20 ± 2.23
**Antioxidant Enzymes**				
SOD (IU/mg protein)	13.86 ± 1.03	10.15 ± 1.04	9.06 ± 1.21	7.84 ± 3.21
CAT (IU/mg protein)	0.07 ± 0.02	0.04 ± 0.01	0.02 ± 0.01	0.01 ± 0.06
GSH (µg/mg protein)	22.12 ± 6.54	17.11 ± 4.32	12.12 ± 3.32	9.13 ± 2.71

IAA = Indole acetic acid; ACCD = 1-aminocyclopropane-1-carboxylate deaminase; EPS = Exopolysachharide; SOD = Superoxide dismutase; CAT = Catalase; GSH = Glutathione reductase. + = present − = absent, ++ = positive, +++ = strong positive, Nd = not detected, % SU = % siderophore units. Values are the average of five replicates and were analyzed by one-way ANOVA followed by Turkey’s test.

## Data Availability

All the data is included in the manuscript file.

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
