# Peer review of "Production of Plant Beneficial and Antioxidants Metabolites by Klebsiellavariicola under Salinity Stress"

_molecules, 2021, doi:10.3390/molecules26071894_

Round 1

Reviewer 1 Report

  • An in vitro experiment is missing. This experiment can easily prove the effect of all the growth-promoting traits. To determine their positive plant growth effect of Klebsiella variicola strain SURYA6, the bacterium has to co-cultivated with Arabidopsis thaliana seedlings (or Triticum seedlings) on Petri dishes in a growth chamber under salt stress conditions and normal.
  • I recommend English editing, professional or by a native speaker.
  • Other recommendations:

Line 21 “keep a check” it is better “can control phytopathogens”

Line 26 “bioinoculant-cum salinity bio-elevator” it is better “bioinoculant under salinity conditions”

Line 69 “helps in the good growth of plant roots” it is better  ”promotes root growth”

Lines 86-88 you should add one more sentence with information about your study, as a more complete epilogue to your introduction

Line 99 two dots, two “was”

Lines 100-101 you should focus in this paragraph on phytase screening medium and then in “2.2.2. Phosphate Solubilization and Organic Acid Production”, mention on line 113 that you screened on Pikovskaya’s and KB agar o.1 ml from the same dilution as phytase medium to grow.

Lines 228-242, paragraphs 3.3.1 and 3.3.2 should be one. These results are connected and they have complementary effects.

Line 253 produced

Lines 250-254 describe the result without all these numbers, they are written in table 1.

Line 217-218, 298, and in Fig 4 legend, mention which phylogenetic tree program you used (program title, version)

Author Response

Authors’ Response to Reviewer 1 Comments

  • An in vitro experiment is missing. This experiment can easily prove the effect of all the growth-promoting traits. To determine their positive plant growth effect of Klebsiella variicola strain SURYA6, the bacterium has to co-cultivated with Arabidopsis thaliana seedlings (or Triticum seedlings) on Petri dishes in a growth chamber under salt stress conditions and normal.

Authors’ Response: Agreed to the comment of the Reviewer. The addition of in-vitro plant experiments will double the size of this MSS. We have already performed in-vitro experiments with T.aestivum and maize under salt stress and normal conditions and have prepared a separate manuscript of 20+ pages on this wider aspect. The said manuscript was submitted to Plants (manuscript id plants-1018737.

  • I recommend English editing, professional or by a native speaker.

Authors’ Response: MSS is duly edited for the English language by native English speaking Co-author (FHH)

Other recommendations

  • Line 21 “keep a check” it is better “can control phytopathogens”

Authors’ Response: The abstract is shortened as per the suggestion of Reviewer 2 and hence the sentence … “keep a check”… is deleted during the shortening

  • Line 26 “bioinoculant-cum salinity bio-elevator” is better “bioinoculant under salinity conditions”

Authors’ Response: Agreed and Revised (Line 29)

  • Line 69 “helps in the good growth of plant roots” it is better  ”promotes root growth”

Authors’ Response: Agreed and Revised as per the suggestion (Line 73)

  • Lines 86-88 you should add one more sentence with information about your study, as a more complete epilogue to your introduction

Authors’ Response: Agreed and additional information about the present study is added in the introduction (Line 92-97).

  • Line 99 two dots, two “was”

Authors’ Response: Extra full stop (.) is deleted (Line 108)

  • Lines 100-101 you should focus in this paragraph on phytase screening medium and then in “2.2.2. Phosphate Solubilization and Organic Acid Production”, mention on line 113 that you screened on Pikovskaya’s and KB agar o.1 ml from the same dilution as phytase medium to grow.

Authors’ Response: Agreed and the text is revised (Line 108,109, 112, 124). 

  • Lines 228-242, paragraphs 3.3.1 and 3.3.2 should be one. These results are connected and they have complementary effects.

Authors’ Response: Agreed and these two sections (3.3.1 and 3.3.2) are merged as one heading.

  • Line 253 produced

Authors’ Response: This typo is deleted during the revision of the paragraph (Line 262-266).

  • Lines 250-254 describe the result without all these numbers, they are written in table 1.

Authors’ Response: Agreed and the paragraph is revised (Line 262-266).

  • Line 217-218, 298, and in Fig 4 legend, mention which phylogenetic tree program you used (program title, version)

Authors’ Response: Phylogenetic analysis was performed using Molecular Evolutionary Genetics Analysis (MEGA) software version7 (Line 227-228)

Reviewer 2 Report

SUMMARY

Manuscript no. molecules-1131372 reports results on the isolation and characterization of several plant growth-promoting bacteria from wheat rhizospheric soil. Based on in vitro traits investigation, among 34 isolates, Klebsiella variicola was selected as a putative biostimulant agent.

The experiments were conducted with valid methodologies, the findings are reported clearly and a valid discussion section has been presented in the perspective of the working hypothesis. Nevertheless, the context centred in the article falls little within the scope of the journal. The authors highlighted only the potential application of the strain as a biostimulant. In my opinion, they should enhance the characterized molecules and extend the discussion for industrial wider purposes. Moreover, concerning the biostimulant potentialities, I think that the authors should provide additional in planta experiments to demonstrate the real potential of this strain to be used as a biostimulant agent. Finally, all the manuscript needs a revision of the English language for many typos and several grammatical errors. Please, see comments for other suggestions.

COMMENTS

Title: The title should be revised for clarity and significance.

Abstract: Should be shortened

Introduction: The introduction correctly places the study in the biostimulant context and highlights why it is important. However, I think that the authors should introduce the exploitation of bacteria for industrial purposes to meet journal aim. Authors should add the working hypothesis in the Introduction section to present better the objectives of the work.

Results: Results are concise and clear. However, I think that additional experiments are needed, i.e. wider strains characterization and in planta experiments.

Discussion: The findings and their implications should be discussed in the broadest context possible, not only related to biostimulant application. Moreover, the limitations of the work are poorly highlighted.

Materials and Methods: The methods are described in sufficient details. The version of the statistical software used should be reported (L224-225).

Author Response

Authors’ Response to the comments of Reviewer 2

Comments and Suggestions for Authors

  • The title should be revised for clarity and significance.

Authors’ response: The title has been revised

  • Abstract: Should be shortened

Authors’ response: Abstract has been shortened from 303 words to 216 words

  • Introduction: The introduction correctly places the study in the biostimulant context and highlights why it is important. However, I think that the authors should introduce the exploitation of bacteria for industrial purposes to meet journal aims. Authors should add the working hypothesis in the Introduction section to present better the objectives of the work.

Authors’ response: Agreed. The working hypothesis and objectives of the study are added (Line 88-97).

  • Results: Results are concise and clear. However, I think that additional experiments are needed, i.e. wider strains characterization and in planta 

Authors’ response: Agreed to the comment of the Reviewer. The addition of in planta experiments will double the size of this MSS. We have already performed in-vitro experiments with T.aestivum and maize under salt stress and normal conditions and have prepared a separate manuscript of 20+ pages on this wider aspect. The said manuscript was submitted to Plants (manuscript id plants-1018737).

  • Discussion: The findings and their implications should be discussed in the broadest context possible, not only related to biostimulant application. Moreover, the limitations of the work are poorly highlighted.

Authors’ response: The finding of the present study have been discussed in the broadest context possible way (Line 328-330,354, 371-373, 434-437).

Limitations of the work are now added (Line 437-439)

  • Materials and Methods: The methods are described in sufficient detail. The version of the statistical software used should be reported (L224-225).

Authors’ response: The version of the statistical software and graph software are now added (Line 236-237).

Round 2

Reviewer 1 Report

The authors provided the presentation quality of the revised manuscript “molecules-1131372-peer-review-v2”. The overall merit is satisfactory.

My suggestion is that the article can be published in the present form.

Reviewer 2 Report

The authors answered correctly all my previous comments. I have no further suggestions.